# Investigation of the Relationship between Vibration Signals Due to Oil Impurity and Cavitation Bubbles in Hydraulic Pumps

Özgür Yılmaz [1,*] , Murat Aksoy [1] and Zehan Kesilmiş [2]

1   Department of Engineering and Architecture, Çukurova University, Adana 01330, Turkey; aksoy@cu.edu.tr
2   Department of Engineering and Architecture, Adana Science and Technology University, Adana 01250, Turkey; zkesilmis@adanabtu.edu.tr
*   Correspondence: ozguryil@gmail.com

**Abstract:** Although hydraulic pumps are frequently used in daily life, improper use due to oil analysis or oil contamination is ignored. There is no instantaneous inspection; instead, the oil is changed periodically at certain times, whether it is contaminated or not. Hydraulic systems operate based on Pascal's law, which states that the fluid will distribute the pressure equally to every point in a closed area. The fluid oil taken from an oil reservoir is moved into the pump by engine power. During this movement, as it passes through different pressure areas and different sections, undesirable events such as viscosity change and gas formation occur in the hydraulic oil. These formations collide with the outer walls and cause cavitation with respect to unwanted oil impurities. This cavitation causes unwanted vibration signals to occur in the normal working order of the system. As a result of cavitation, the particles that affect the lubricity and fluidity of the oil in the oil are mixed into the liquid and circulate freely. At the connection points, the blockage caused by the liquid in the pump cylinder block or the valve plate and the collisions of particles is effective. As a result, it creates vibrations of different frequencies. The frequency and amplitudes of these vibrations differ according to the degree of oil contamination. A method has been developed to find the degree of contamination of the oil circulating in the pump by looking at the amplitude and frequency of these vibrations measured from the motor body. There exist standards about the pollution of hydraulic fluid. With these standards, the maximum number of particles allowed for a given pollution level is defined. This topic is discussed in the conclusion to this study. This method has also been proven experimentally. Error and vibration analysis studies on pumps using a different approach are available in the literature. In these studies, pressure variation, total energy transmission, or artificial intelligence models were used to detect anomalies in the pump. In this study, the impurity rate of the oil was set at five different levels and the operating regime of the pump at each level was investigated experimentally. Rayleigh–Plesset and Zwart–Gerber–Belamri models, which are the most common cavitation models, were used to explain the bubble formation in the moving oil and the relationship of these bubbles with vibration. Frequency components were examined by the Discrete Fast Fourier Analysis method, where the operation of the pump was affected by the increase in oil impurity.

**Keywords:** pump; hydraulic pump; vibration; oil; cavitation; oil impurity; cavitation bubbles

## 1. Introduction

In today's world, hydraulic systems are present in almost every sector. The components that make up these systems (pump, valve, hose, fittings, etc.) play a crucial role. However, the most important component that causes malfunctions in hydraulic systems is oil. The total system characteristic starts to change as the oil loses its properties, with loss of efficiency, variable amplitude, and vibration signals observed in the characteristic. As a consequence of these changes, premature failures may occur [1]. However, there is no active method to prevent or purify the oil contamination in a working hydraulic system [2].

The hydraulic pump is the mechanism that converts mechanical energy into hydrostatic energy. With a method based on the principle of volume change with the rotational movement obtained from an engine, it starts the flow towards the moving parts by absorbing the fluid from the tank. The drive system that provides the rotational movement can be an electric motor or an internal combustion engine. The motor used in this study is an electric three-phase asynchronous motor. There are different types of hydraulic pumps available, with a gear hydraulic pump used in this study.

In hydraulic systems, the fluid circulating in the system is exposed to different effects. The passage of different pressure regions, temperature changes, humidity, and the total number of acids and bases of the fluid affect the viscosity of the fluid and the total system performance. If the molecules consisting of carbon (C) and hydrogen (H) atoms in the oil combine with oxygen (O) due to the effect of water, carboxyl (COOH) group acids form. The different acids formed produce abrasive and corrosive effects on the surfaces where the oil works. In this instance, the parts detached from the surface mix with the oil again and circulate in the system and show impact and compression effects at different points, such as the narrowing of the flow path and coupling points. These effects create vibration.

Oil pollution is not the only effect that creates vibration. It is possible that many effects such as looseness of the connection points, fastening strength, and misalignment can cause vibration. However, in this study, these effects were kept constant and only the oil pollution levels were changed and the difference in the resulting signs was examined.

Hydraulic pumps do not create pressure. This pressure is due to the resistance against the flow of the fluid. When the fluid's aforementioned flow parameters change, the pump's operating state also changes.

Health monitoring of hydraulic pumps is currently achieved through oil analysis [3]. Oil performance degradation causes corrosion, carbon formation, the loss of oil lubricity, filter clogging, and the elevation of local temperatures on friction surfaces, thus accelerating the wear of friction units [4,5]. The online monitoring of the condition of the lubricant makes possible not only its more effective usage but also the prevention of mechanism failure [6]. Conventional signal processing techniques, including time and frequency domain analysis, are based on the assumption that processing signals are stationary and linear [7].

There are published articles and suggested methods on error analysis of hydraulic pumps. Inputs such as pressure, energy, temperature, voltage, current, and rotation speed were used as data inputs in the models frequently encountered in these studies. In the methods used, artificial intelligence, pulsation, wavelet model, and similar methods were found. However, unlike the methods in the literature, the vibration model caused by the problems that occur in the system during the flow of hydraulic oils is not associated. The method in this study reveals that with an increase in the number of bubbles formed by cavitation, the effect of the pushing force on the solid surface created by the particles removed from the surface increases in total along with the vibration created by both axes.

Harihara and Parlos tried to detect pump faults by changing motor electrical signals [8].

Siano and Panza used a centrifugal pump when working on the analysis of vibration signals, and attempted prediction with a nonlinear auto-regressive model [9].

Gao et al. proposed a pulsation model on the error analysis of hydraulic pumps. They worked on the wavelet analysis method in the model in which the pumps make use of inlet and outlet pressure differences [10].

Kamiel also worked on vibration-based multi-fault diagnosis using vibration signals. A PCA model was investigated for this topic, with the proposed PCA method transforming the original variables (30 features) into 30 principal components that are uncorrelated [11].

Chen et al. developed a genetic algorithm by using an adaptive wavelet transform of the vibration signals on a water hydraulic pump [12].

However, a data-driven approach is suggested by Casoli et al. for fault identification and classification in hydraulic axial piston pumps [13].

Muralidharan and Sugumaran proposed a J48 algorithm using discrete wavelets for error diagnosis. This study was carried out for monoblock centrifugal pumps [14].

Al-Badour et al. used a wavelet technique for fault diagnosis of a rotating machine. They proposed a hybrid model using wavelet analysis with a Fourier-based time–frequency analysis approach [15].

Hamomd presented a study on the analysis of modulated signals. The proposed method MSB (Modulation Signal Bispectrum) analysis technique is used to extract these deterministic features of modulation. Components in the low-frequency band are used to diagnose both bearing defects and impeller blockages [16].

By relying on the above publications' reviews, it can be said that it is not possible to handle the cavitation mathematical modeling and analysis with FFT to analyze the diagnosis of the vibration changes on the hydraulic pump [17]. In this study, the vibration signals received from the hydraulic pump were measured and then analyzed by FFT. Analysis results are given in a filtered manner in frequency change. According to Fourier theory, signals can be expressed as a sum of sine waves of different phases and frequencies, no matter how complex they are [18].

Many publications in the literature discuss invasive methods based on fluid and mechanical measurements. Most publications on this subject make error diagnoses with different methods and discuss developed methods for diagnosing the error that occurred. The method presented in this study is an association method that allows the traceability of situational changes. With this method, the formation of new frequency components in vibration signals, the formation of sidebands, and spectral leakages can be observed and information about the state of the system fluid can be obtained. As a consequence, a total performance assessment of the system can be exhibited. With the time–frequency analysis and oil pollution association method presented in this study, the operating regime of a hydraulic system and the criteria for an oil change can be easily monitored with a single indicator. Estimation can be made for oil impurity or predictive maintenance.

Vibration signals are a very powerful prognostic imaging tool, not only for diagnosis but also for condition monitoring systems. It has been stated in the literature that the pump operation and the vibration signals taken from it are evaluated as a fault diagnosis and can be classified [16]. However, many factors affect the operation of the system. In addition to mechanical reasons such as connection type, misalignment, and pollution at the pump outlet, analog data such as temperature, pressure, and humidity are also factors affecting the operation of the system. There are many factors such as oxidation and the presence of water and dissolved air that also affect oil pollution [17]. However, the subject of this study is the examination of pollution caused by cavitation.

## 2. Effect of Dirty Oil

Oil contaminants are the key factors impacting hydraulic system service life. Contamination in hydraulic systems can be classified into particle contaminants (metal particles from wear or dirt ingression) or chemical contaminants (water, air, acid number, etc.). This contamination effect increases the corrosive behavior with increases in temperature. Examples of damage from contamination include accelerated component wear, orifice blockage, formation of rust or other oxidation, depletion of additives, formation of other chemicals, and oil degradation [13]. Three types of these contaminants (solid particles, water, and gas) have been identified as the primary issues for hydraulic component life [19]. Failures arising from contamination falls into three categories:

- Catastrophic failure;
- Intermittent failure;
- Degradation failure.

The main reason catastrophic failure occurs is large particles. When a large particle enters the pump or valve, it causes a jam and the engine shuts down. Hence, it results in motor seizure. While the reason provided above describes catastrophic failure, intermittent failure occurs when a poppet valve or spool valve is contaminated. This contamination

results in sealing error. Degradation failure emerges as a result of contamination erosion, corrosion, and wear. An increase in internal leaks will cause total system failure over time [19].

Particles in the oil may have been formed or moved by many effects. These are usually caused by effects occurring in the chamber or are carried into the chamber by a carrier such as (or similar to) air. Since these particles are smaller than 14 μm in size, they cannot be seen. They can be detected with a microscope or by analyzing the system's unusual behavior, such as vibration [20].

Water is a fatal chemical contaminant for hydraulic systems. The presence of the water compound affects many aspects of the oil. As a result, the jet effect can accelerate and the moisture level of the oil, in other words, may cause oil molecules to slow down. Corrosion of the oil reservoir, the jet effect at the corner points, and erosion due to cavitation can indicate the excess water component. These effects may have various slippery reducing effects due to the compression of water crystals at low temperatures. Excessive water content may result in additive depletion, oxidation, and acid, alcohol, or sludge effects. The amount of water dissolvable in the oil is called the saturation level, and if water is soluble above this level, the appearance of the oil becomes cloudy. The amount of water indicating the saturation of oil is typically 200–300 ppm at 20 °C [21].

Air may exist in oil, either dissolved or free. Dissolved air in oil does not pose a problem as long as it remains in solution. However, undissolved air may generate heat under pressure. The air heated with high pressure may cause an increase in oil temperature and thus foaming in the oil chamber.

Overheating may cause oil molecules to chemically change form and the additive material to run out. In general, there are four main sources of contamination in hydraulic oil.

- Native contamination;
- Contaminated new oil;
- Ingressed contamination;
- Internally generated contamination.

Some contaminant particles may have infiltrated the oil during system operation or maintenance. Materials such as welding slag, machining swarf, and excessive sealant that mix into the oil from the outside are examples of native types of contaminations.

Water, moisture, and air components resulting from additives in the oil are under the influence of high pressure over time depending on the operating conditions. Components hitting the surface with this pressure cause damage to the surface. This causes more additives to be released and therefore more surface collision. This is called a wear regeneration cycle.

In general, a hydraulic system fails due to the contamination of the oil in the fittings and transition elements. Blockages in pumps refer to the situation in which the pumped liquid contains materials such as rags, fibers, wood chips, abrasive sand, sand, or bricks that can wrap around the impeller or choke the area between the impeller blades to prevent or even stop rotation.

Studies have shown that oil pollution has many effects, with one of the biggest being cavitation. Cavitation has been shown to cause vibration at different frequencies [22]. For this reason, it is necessary to understand the mathematical model in order to examine the frequencies caused by cavitation.

## 3. Cavitation Phenomenon

Cavitation is a hydrodynamic phenomenon that occurs as a result of the evaporation of fluid in a low-pressure region. Air particles, particles, and different factors in the fluid affect the formation of cavitation. When the vapor particles formed reach the higher-pressure region, they liquefy again. However, during this event, pressure pulses occur which causes vibration. Researchers have revealed that these vibrations increase with the increase in the number of particles in the liquid [23]. Therefore, the higher the oil impurity, the higher the

vibrations due to cavitation. The amplitude of the resulting vibrations is the major indicator of oil impurity. Therefore, the variation and amplitude of the frequency components in the vibrations will increase with the impurity of the oil.

Cavitation is caused by the formation of vapor-filled bubbles and their sudden bursting. The cavitation problem usually occurs as a result of the pressure in the pump falling below the vapor pressure of the pumped fluid. Cavitation damage is a threat to the pump. To prevent cavitation, there must be a minimum pressure at the suction port, defined as the net positive suction head (NPSH), so that the liquid does not boil or evaporate. Due to this minimum pressure, cavitation formation can be prevented. The pressure applied at the suction port must be higher than the vapor pressure of the fluid. If there is cavitation damage in the pump, it may be necessary to throttle the control valve located on the pressure side. It is possible to reduce the flow rate and the NPSH value required by the pump by throttling the control valve. The important point is to ensure that the residual flow is high enough to cool the pump.

As a result of the cavitation problem, small gas bubbles form in the liquid. These bubbles reduce the capacity of the pump over time and thus shorten its life. Cavitation also causes the pump to function noisily, which is one of the most disturbing situations. It occurs as a result of the rapid movement of any substance in a fluid. We can also define cavitation as a type of phase change event.

Cavitation is valid for all systems with pressure and velocity changes, even in the human body. Joint problems are common. Cavitation is directly proportional to the ambient temperature. With the decrease of the evaporation temperature, a cold boiling occurs in the system with water vapor and air bubbles, and this event leads to cavitation. The lowest pressure in a pump occurs at the inlet of the impeller; therefore, it is possible to say that cavitation in pumps is usually collected at the inlet of the impeller.

Cavitation is dependent on pressure changes, but it is a preventable problem by controlling the pressure decrease. It shortens the life of the pump, causes noise in the pump, and causes wear and abrasive effects. This abrasive wear and surface fatigue is responsible for about 90% of degradation failures [24].

For this reason, examining the factors that cause cavitation with a model is a more result-oriented approach.

### 3.1. Cavitation Influencing as a Source of Vibration

Many mathematical models have been developed for cavitation. One, the Zwart–Gerber–Belamri (ZGB) cavitation model, derived from the Rayleigh–Plesset (R–P) equation, is commonly used in cavitation calculations [25]. This method gives suitable results in the numerical calculation of cavitation in the presence of very small particles in a fluid. In addition, the density of particles in the fluid does not excessively affect the calculations. This dynamic cavitation relation, derived from the Rayleigh–Plesset equation, can be expressed as follows:

$$R_B \frac{d^2 R_B}{dt^2} + \frac{3}{2} \left( \frac{dR_B}{dt} \right)^2 + \frac{2T}{R_B} = \frac{p_v - p}{\rho_l} \tag{1}$$

Here, $R_B$ is the radius of cavitation, T is the surface tension coefficient, $p_v$ is the evaporation pressure, and $\rho_l$ is the density of the liquid. The size of the particular bubble can be calculated by using the generalized Rayleigh–Plesset equation as follows [26]:

$$\frac{p_v(t) - p_\infty(t)}{\rho_l} = R \frac{d^2 R_B}{dt^2} + \frac{3}{2} \left( \frac{dR}{dt} \right)^2 + \frac{4V_l}{R} \frac{dR}{dt} + \frac{2T}{\rho_l R} \tag{2}$$

Here, $\rho_l$ is the liquid density, $R$ is the bubble radius, and $V_l$ is the kinematic viscosity of the fluid. Liquid pressure values and bubble pressures are known. Blake and Neppiras et al. showed that a critical radius exist which determines the stability of bubbles [27]. A bubble is stable if its radius is smaller than the critical radius; if its radius is bigger than the critical

radius, it will grow very fast. This is referred to as the Blake critical radius. However, a threshold pressure for the bubble can be derived from the Blake critical radius. This means that a pressure below the critical pressure will cause the balloon to grow rapidly [28]. As the bubbles move through the pump with the liquid, they reach a higher-pressure area and burst rapidly. It is well known that the noise that occurs when bubbles burst is due to a sudden increase in pressure. This effect causes the relaxation of a huge amount of concentrated energy, which can be very destructive. If a bubble bursts near a solid surface (in an impeller blade or curved body), a shockwave appears that strikes and damages the surface. This continuous process leads to pump material conveyance, resulting in increased surface roughness and, ultimately, a hollow impeller. This results in unsuitable flow conditions and lower efficiency, and also increased noise and vibration [27].

For the practical modelling of cavitation, the equation can be simplified by neglecting the surface tension, viscosity, and second-order time variable. As a result, the frequency and amplitude of the vibration changes. The quadratic term can be simplified as follows:

$$\frac{dR_B}{dt} = \sqrt{\frac{2}{3}\frac{p_v - p}{\rho_l}} \tag{3}$$

Zwart et al. used the total interphase mass transfer rate by calculating the bubble density numbers for each volume in the ZGB cavitation model [28]. Here, the interphase mass transfer $R$ is denoted by:

$$R = n_o \left( 4\pi R_B^2 \rho_v \frac{dR_B}{dt} \right) \tag{4}$$

The number of cavitations ($n_o$) per unit volume depends on the direction of the phase change. The development of cavitation is expressed by the following equation.

$$n_o = (1 - \alpha_v)\frac{3\alpha_{ruc}}{4\pi R_B^3} \tag{5}$$

With a default value of $\alpha_{ruc}$, the disappearance of cavitation, i.e., condensation, is indicated by:

$$n_o = \frac{3\alpha_v}{4\pi R_B^3} \tag{6}$$

When the above relations are arranged, the cavitation equation has the form:

$$R_e = F_{cond}\frac{3\alpha_{ruc}(1 - \alpha_v)\rho_v}{R_B}\sqrt{\frac{2}{3}\frac{p_v - p}{\rho_l}} \quad \text{if } p < p_v \tag{7}$$

$$R_c = F_{vap}\frac{3\alpha_v \rho_v}{R_B}\sqrt{\frac{2}{3}\frac{p_v - p}{\rho_l}} \quad \text{if } p > p_v \tag{8}$$

Here, $\rho_v$ is the vapor density, $\rho_l$ is the liquid density, $\alpha_v$ is the vapor volume ratio, and $F_{vap}$ and $F_{cond}$ are two empirical correction coefficients corresponding to the evaporation and condensation process, respectively. $F_{vap}$ and $F_{cond}$ are not equal, as the condensation process is usually much slower than evaporation [18]. If $p < p_v$ it can be termed a vapor production, while if $p > p_v$ it can be termed a vapor destruction. The source term of vapor production and vapor destruction is expressed by $R_e$ and $R_c$, respectively. This equation contains four parameters and if default values are used here, an equation is obtained depending on the density values of $\rho_v$, $\rho_l$, and $\alpha_v$, and each other. This defines a situation where cavitation is affected according to the density change and changes according to the $p$ and $p_v$ changes. The defined default values are as follows: $R_B = 2 \times 10^{-6}$ m, $\alpha_{ruc} = 5 \times 10^{-4}$, $F_{vap} = 50$, and $F_{cond} = 0.01$. As can be seen from this equation, since particles or other substances in the oil will change the oil density, cavitation formation will

also be affected. It was observed that cavitation increased as the number of particles in the fluid increased [29]. It has even been determined that turbulence occurs in the fluid in some cases. This turbulence also increases vibration. Therefore, the most important factor affecting the formation of vibrations in the pump is cavitation. In this case, the degree of cavitation, hence the degree of impurity, can be determined by measuring the vibration in the system.

The frequency of the pressure pulses that occur during cavitation depends on the angular cogwheel structure of the pump, the type of fluid, and the flow rate. Depending on the turbine's rotational frequency, these pressure pulses can contain many frequency components. One method used to detect cavitation is to measure the noise level created by vibration [22]. However, in this method, outdoor noise significantly affects the measurement accuracy.

As a result, the total effective value of vibrations at different frequencies will show the level of cavitation. For this, an experimental system was designed in which vibration amounts were measured in two directions of the *X*-axis and *Y*-axis. In this system, vibrations on an oil pump were measured with sensitive MEMS-based sensors. Depending on the level of contamination of the oil data, the variance of the measurement results was transferred to the computer environment with the help of a specially designed interface. A Discrete Fourier Transform was used to calculate the frequencies and amplitudes of the vibrations by using data taken from the total system.

### 3.2. Cavitation Bubble Dynamics

When discussing the phenomenon of cavitation, it is necessary to mention the pressure created by the bubbles forming the cavitation on the solid surface and the movements of these bubbles in Figure 1.

As a result of these movements, the disassembled parts due to the pressure formed on the inner walls of the pump and the gas that created the internal pressure of the bubbles was replaced [30]. The cavitation equation was previously introduced above with the Rayleigh–Plesset and ZGB models, which are regarded as the most famous models to describe the evolution and collapse of spherical symmetrical bubbles in liquid.

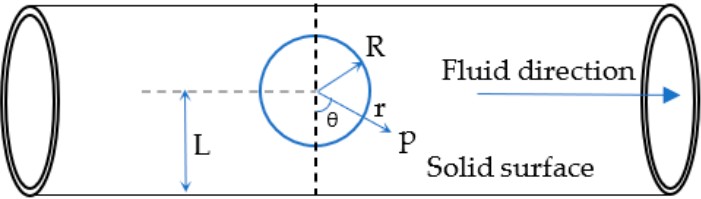

**Figure 1.** Bubble movement and pressure angle [31].

The bubbles do not have to be uniform, but here it is assumed that they move by retaining their spherical shape. As a bubble moves throughout the pump on a solid uniform boundary and gravity, heat transfer and gas diffusion effects are negligible.

Based on the assumptions above, the velocity equation of the bubbles can be expressed as [31]:

$$\frac{1}{r^2} + \frac{\partial}{\partial r}\left(r^2 \frac{\partial \Phi}{\partial r}\right) + \frac{1}{r^2 sin\theta}\frac{\partial}{\partial \theta}\left(sin\theta \frac{\partial \Phi}{\partial \theta}\right) = \frac{1}{c^2}\frac{\partial^2 \Phi}{\partial t^2} \tag{9}$$

Here, *r* is the distance from the bubble center, *c* is the speed of the sound at infinity, Φ is velocity potential, and *t* is time.

Under negative pressure conditions, bubbles occur and grow. As the pressure increases, the cavitation bubble is compressed to a small radius, releasing large shock pressure up to tens of GPa penetrating the solid boundary. This shock pressure affects the solid boundary with respect to the ratio between the distance and the maximum bubble radius. The shock pressure caused by the bubble bursting is an important cause of cavitation erosion,

especially for micrometer radius bubbles. Shock pressure can be calculated using this equation:

$$P_{collapse} = P_\infty + (P_\infty - P_v) \frac{\left(\frac{R_0{}^3}{4R^3} - 1\right)^{\frac{4}{3}}}{\left(\frac{R_0{}^3}{R^3} - 1\right)^{\frac{1}{3}}} \tag{10}$$

where, $P_\infty$ is the liquid pressure at infinity, $P_v$ is a vapor pressure, $R$ is the bubble radius, and $R_0$ is the bubble radius at a certain time.

The pump's solid boundaries change the waveform of the pressure on the liquid due to this vibration. The change of the pressure waveform also affects bubble formation and the progression of this process. The relationship between the timing of bubble formation and collapse, the distance of bubbles from solid boundaries, and their velocity with vibration has been presented in the literature [31].

## 4. Examination of the Vibration Frequency Spectrum

The vibration signals represent the sum of the harmonic components in the form of the fundamental frequency component and its multiples. As a result of the degradation of the oil used over time, there are changes in the amplitude of the frequency components in the vibration signals. The most suitable solution to detect these frequency components and changes is the use of a Discrete Fourier Transform (DFT) [17,18].

In order to apply DFT to vibration signals, signals that change depending on time must be sampled at a certain time interval and converted into discrete time. According to the Nyquist theorem, the sampling time should be less than half the period of the highest frequency component in the vibration signal [32]. For this reason, the sampling frequency was chosen to be high because the value of the greatest vibration frequency in experimental studies was unknown.

The Discrete Fourier Transform is a linear transform that is computed when a series of samples is separated by a single time. If we consider $f(t)$ as the source vibration signal and the number of samples taken are denoted by $f[0], f[1], f[2] \ldots f[N - 1]$, accordingly, the Fourier transform of the vibration signal is expressed by:

$$F(j\omega) = \int_{-\infty}^{+\infty} f(t)e^{-j\omega t}dt \tag{11}$$

Each sample $f[k]$ can be scrutinized as an impulse which has area $f[k]$. So, integrands of the sample points can be represented as:

$$F(j\omega) = \int_{0}^{(N-1)T} f(t)e^{-j\omega t}dt \tag{12}$$

$$= f[0]e^{-j0} + f[1]e^{-j\omega T} + f[2]e^{-j\omega 2T} + \ldots + f[k]e^{-j\omega kT} + \ldots f[N-1]e^{-j\omega(N-1)T} \tag{13}$$

If we denote this sum with an equation:

$$F(j\omega) = \sum_{k=0}^{N-1} f[k]e^{-j\omega T} \tag{14}$$

these equations are valid for any $\omega$, but since we take $N$ data samples of vibration signals, the function will give $N$ outputs.

If it is thought that we perform the continuous Fourier transform for a finite interval, DFT will be valid for $N$ samples. In this case, $N$ samples taken in discrete time would be expected to have an image similar to the one below.

Here, we can express the components and harmonics as follows, since the signal components we obtain for FFT are periodic.

$$\omega = 0, \frac{2\pi}{NT}, \frac{2\pi}{NT} \times 2, \ldots, \frac{2\pi}{NT} \times n, \ldots \frac{2\pi}{NT} \times (N-1) \tag{15}$$

Here, $\frac{1}{NT}$ Hz, $\frac{2\pi}{NT}$ rad/s is one cycle per sequence fundamental frequency. We can represent this in a general equation:

$$F[n] = \sum_{k=0}^{N-1} f[k] e^{-j\frac{2\pi}{NT}nk} \tag{16}$$

Here, $F[n]$ is a DFT of the sequence $f[k]$.

As a consequence, the equation can be written in a matrix form as follows:

$$
\begin{pmatrix} F[0] \\ F[1] \\ F[2] \\ . \\ . \\ . \\ F[N-1] \end{pmatrix}
=
\begin{pmatrix}
1 & 1 & 1 & 1 & . & . & . & 1 \\
1 & W & W^2 & W^3 & . & . & . & W^{N-1} \\
1 & W^2 & W^4 & W^6 & . & . & . & W^{N-2} \\
1 & W^3 & W^6 & W^9 & . & . & . & W^{N-3} \\
. & . & . & . & . & . & . & . \\
. & . & . & . & . & . & . & . \\
. & . & . & . & . & . & . & . \\
1 & W^{N-1} & W^{N-2} & W^{N-3} & . & . & . & W
\end{pmatrix}
\cdot
\begin{pmatrix} f[0] \\ f[1] \\ f[2] \\ . \\ . \\ . \\ f[N-1] \end{pmatrix}
\tag{17}
$$

Here, $W = \exp\left(-2j\pi/N\right)$ and $W = W^{2N}$, etc. $= 1$ [17].

## 5. Experimental Studies

In order to examine the effect of oil quality on a hydraulic pump system, an experimental setup consisting of one 80 Bar hydraulic pump, one 1.1 kW three-phase asynchronous motor, and a motor drive was designed. The block diagram of the test setup is depicted in Figure 2.

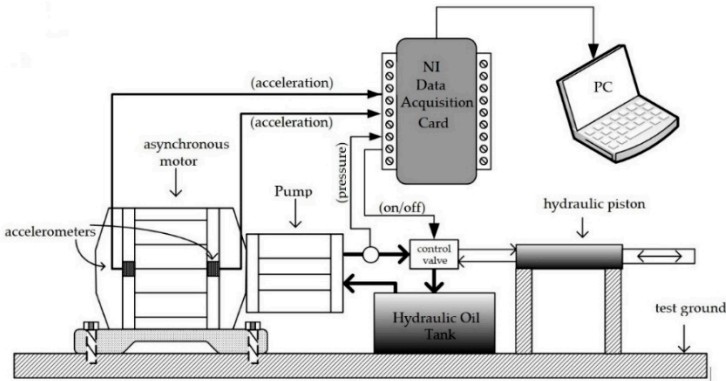

**Figure 2.** Experimental setup to examine the effect of oil quality on a hydraulic pump system.

The measurement results obtained were analyzed using the proposed FFT analysis, with the results depicted in Figures 3–6.

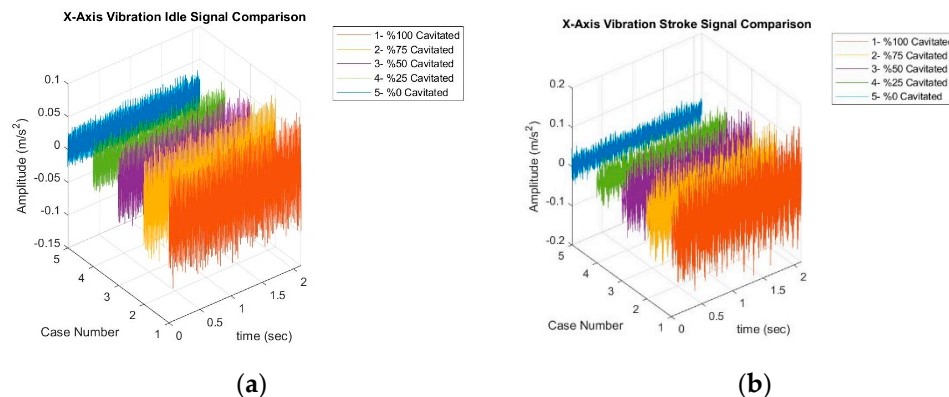

(**a**)       (**b**)

**Figure 3.** Contaminated oil vibration characteristic of (**a**) *X*-axis idle and (**b**) *X*-axis stroke.

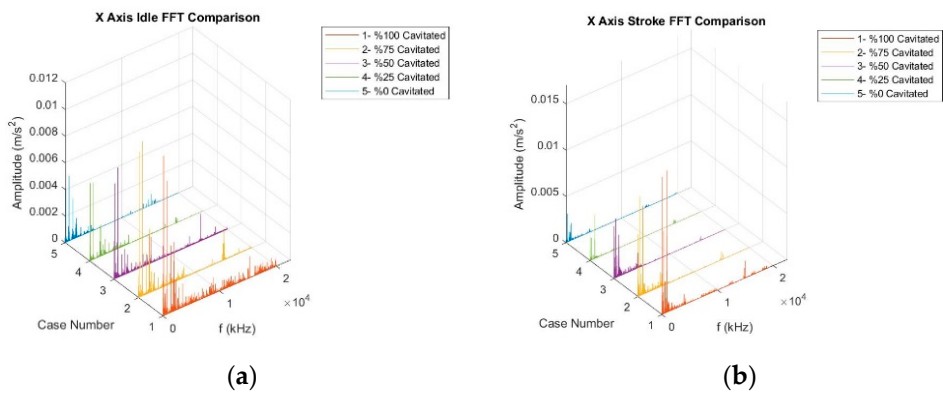

(**a**)       (**b**)

**Figure 4.** Contaminated oil FFT responses of (**a**) *X*-axis idle signal and (**b**) *X*-axis stroke signal.

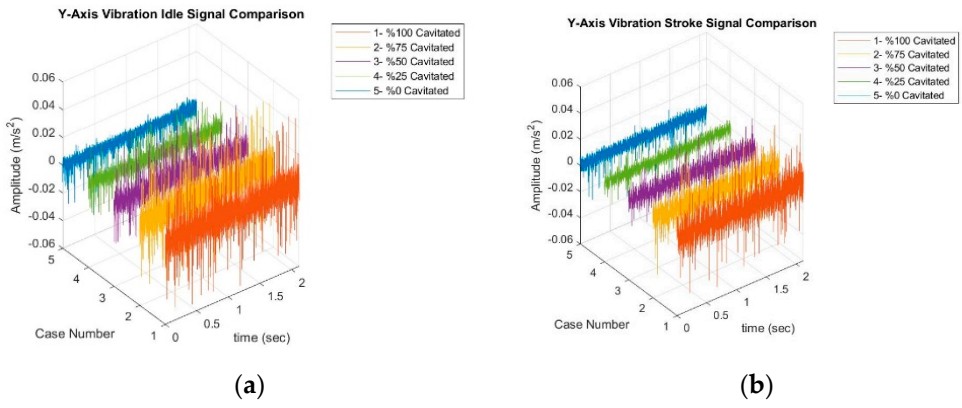

(**a**)       (**b**)

**Figure 5.** Contaminated oil vibration characteristic of (**a**) *Y*-axis idle and (**b**) *Y*-axis stroke.

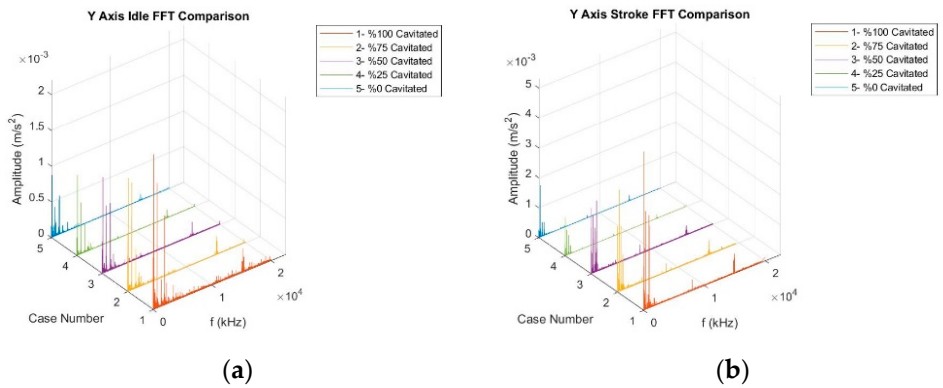

(**a**)       (**b**)

**Figure 6.** Contaminated oil FFT responses of (**a**) *Y*-axis idle signal and (**b**) *Y*-axis stroke signal.

As can be seen in Figure 3, a vibration occurs on the motor body when the system starts to work. The amplitude of this vibration increases as the impurity of the oil is increased. This is a 2 s idle vibration graph. Figure 4 shows the vibration signals from the stroke of the pump 2 s after the motor is energized. Here, too, the increase in the amplitude of the vibration signals and the change according to the impurity state of the oil can be seen. It has a higher amplitude in the stroke movement than the vibration values during idle running time. This reflection can also be seen in the frequency spectrum.

With the experimental setup, measurements were made in five different situations. These measurements show five impurity degrees of contaminated oil in the hydraulic tank: 0%, 25%, 50%, 75%, and 100%. The measurements were taken on the main body of the asynchronous motor working to transfer the oil into the tank.

### 5.1. Limitations of Experimental Setup

The aim of the study is to investigate the vibration realized by cavitation in the frequency domain. For this reason, active values of oil contamination have been changed. The measurement setup and information about the working conditions of the system are described below:

1.  The pump pressure used was 80 bar. This type of pump was chosen because it is one of the most frequently used models in the industry. The pump diameter was 7 cm and the cylinder thickness was 2 cm.
2.  The engine was started for 4 s and stroked at the start of the 2nd s. The system has been operated for a long time. However, since no characteristically different situation was encountered, the working time was analyzed by dividing it into two periods.
3.  In order to reduce the load effect to zero, the system was operated without load.
4.  Conditions are not ideal. For this reason, some parasitic signals were found in the frequency domain.
5.  The experimental setup was fixed on a platform. Vibration that may occur due to the platform and the loose fitting of the system was minimized.
6.  Measurements were taken by measuring displacement (g) with MEMS accelerometers. At the same time, sensor output voltages were measured in DC. All samples were taken on the motor body as *X*-axis and *Y*-axis. For 4 s separately on each axis, 160,000 samples were taken. Vibration signal samples were taken from the front and rear bearings of the engine.

### 5.2. Vibration Signal Frequency Response

The vibration signal was measured for 4 s and this signal was divided into two parts and analyzed. The first 2 s were examined as the idle time of the engine, while the other 2 s were the time that the stroke moved.

Similar changes are experienced in the *Y*-axis, as are the changes in the frequency spectrum against the oil impurity seen in the *X*-axis. With the increase in oil impurity, the amplitudes of the vibration signals increase and, as a result, sideband formation and an increase in vibration frequency amplitudes are observed in the components in the frequency spectrum.

The operating frequency and the components in the frequency spectrum in pumps (briefly, the operating regime of the pump) depend on many factors such as the connection location, looseness, and stroke movement. The frequency spectrum changes according to these component parameters. In our experimental studies, the operating frequency of the hydraulic pump varies between 0–0.5 kHz, according to our measurements. However, the component between about 14–15 kHz varies due to oil pollution. As can be seen, the cavitation vibration signals' frequency component originating from pollution occurs at about 15 kHz [33].

If the results seen in the Figures 7 and 8 which is obtained from the vibration data are compared on a table, the frequency spectrum can be examined with 5 kHz step size and

the largest frequency component amplitude can be examined, so that we can distinguish the operating regime of the pump and the components caused by vibration.

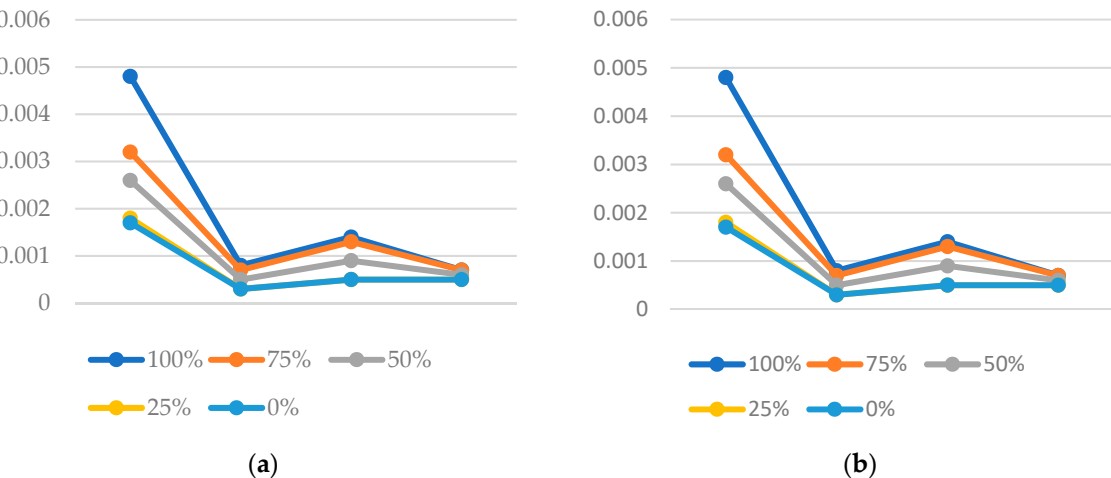

(**a**)

(**b**)

**Figure 7.** Highest frequency amplitudes between 5 kHz steps of *X*-axis (**a**) idle signal and (**b**) stroke signal.

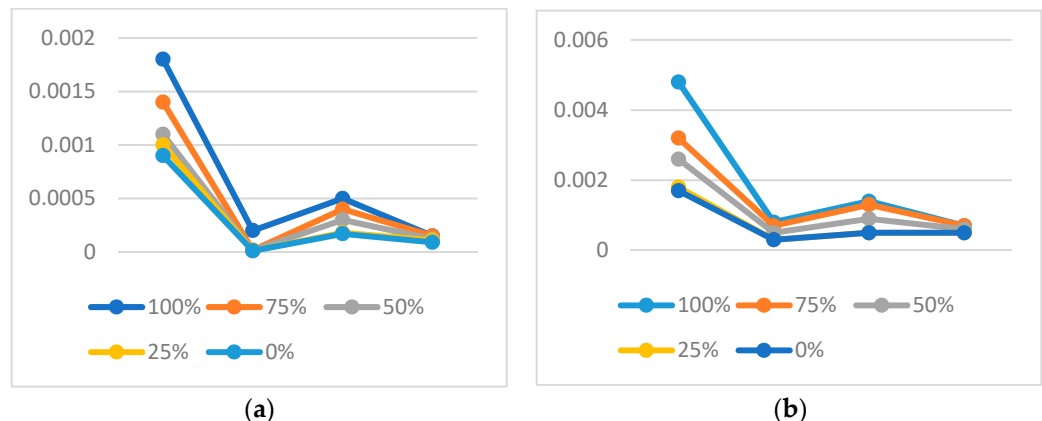

(**a**)

(**b**)

**Figure 8.** Highest frequency amplitudes between 5 kHz steps of *Y*-axis of (**a**) idle signal and (**b**) stroke signal.

All graphs show that there was an increase between 10–15 kHz in addition to the nominal operating regime of the pump and the effect of this component increased proportionally to the oil impurity. It can also be deduced from these tables that the system has a high amplitude natural operating frequency component between 0–5 kHz.

This method is a traditional time–frequency analysis. By observing the relationship between the data obtained with this method and the condition of the oil, it will be easier to prevent malfunctions as pollution can be determined beforehand.

The results obtained here indicate that an increase in oil pollution creates high amplitude frequency components, as shown in the Fourier analysis. These effects can be seen from the data presented in Tables 1 and 2. Amplitudes increasing with pollution and spectral leakages and noise on the floor give visible results.

**Table 1.** Highest frequency amplitudes between 5 kHz steps of *X*-axis.

| Impurity | Idle Signal | | | | Stroke Signal | | | |
|---|---|---|---|---|---|---|---|---|
| | 0–5 | 5–10 | 10–15 | 15–20 | 0–5 | 5–10 | 10–15 | 15–20 |
| 100% | 0.0083 | 0.0016 | 0.0023 | 0.0008 | 0.014 | 0.0025 | 0.004 | 0.001 |
| 75% | 0.0079 | 0.0015 | 0.0020 | 0.0007 | 0.010 | 0.0022 | 0.003 | 0.0008 |
| 50% | 0.0056 | 0.0011 | 0.0017 | 0.0005 | 0.005 | 0.0017 | 0.002 | 0.0006 |
| 25% | 0.0049 | 0.0009 | 0.0014 | 0.0004 | 0.003 | 0.0012 | 0.001 | 0.0004 |
| 0% | 0.0048 | 0.0008 | 0.0012 | 0.0004 | 0.002 | 0.0010 | 0.001 | 0.0003 |

**Table 2.** Highest frequency amplitudes between 5 kHz steps of *Y*-axis.

| Impurity | Idle Signal | | | | Stroke Signal | | | |
|---|---|---|---|---|---|---|---|---|
| | 0–5 | 5–10 | 10–15 | 15–20 | 0–5 | 5–10 | 10–15 | 15–20 |
| 100% | 0.0018 | 0.0002 | 0.0005 | 0.00015 | 0.0048 | 0.0008 | 0.0014 | 0.0007 |
| 75% | 0.0014 | 0.000017 | 0.0004 | 0.00015 | 0.0032 | 0.0007 | 0.0013 | 0.0007 |
| 50% | 0.0011 | 0.000014 | 0.0003 | 0.00012 | 0.0026 | 0.0005 | 0.0009 | 0.0006 |
| 25% | 0.0010 | 0.000012 | 0.00018 | 0.00011 | 0.0018 | 0.0003 | 0.0005 | 0.0005 |
| 0% | 0.0009 | 0.000011 | 0.00017 | 0.00009 | 0.0017 | 0.0003 | 0.0005 | 0.0005 |

*5.3. Results and Discussion*

Bubbles collapse non-symmetrically, forming a liquid microjet that subsequently ruptures resembling a "hammer effect" [34]. A bubble collapse is shown in Figure 9.

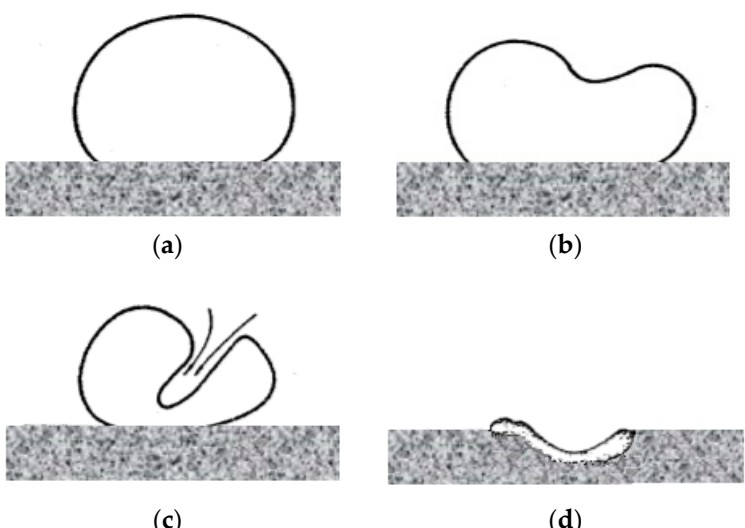

(**a**)            (**b**)

(**c**)            (**d**)

**Figure 9.** Mechanism of cavitation bubble collapse. (**a**) Initial bubble; (**b**) Initiation of bubble collapse; (**c**) Formation of liquid jet; (**d**) Impact and metal extrusion.

The cavitation potential depends on the type of material, especially its tensile strength and initial condition. If the explosions caused by this type of bubble collapse have sufficient impact force, repeated fatigue fracture occurs on the surface. A typical bubble pressure implosion shown in Figure 10. This breaking event also depends on the formation of different values according to the material type. Okada et al. showed that impact loads of 9.1 N (aluminum), 9.7 N (copper), and 13.7 N (mild steel) were required to form a 4 µm pit from a single implosion of a single bubble [35]. However, bubble collapse pressures

will vary with the bubble size, shape, and location. Since steel was used in this study, the expected displacement average value can be evaluated as 13.7 N.

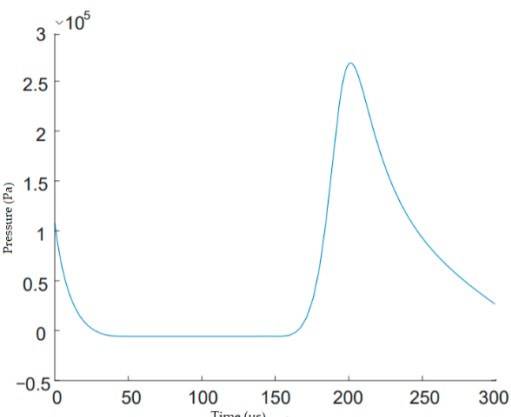

**Figure 10.** Pressure induced to bubbles over time.

The effect that such a force will create on the sensor mounted on the surface can be calculated with the following equation [34]:

$$F(N) = \left(\frac{1}{g_c}\right) \times \left(\frac{s}{t}\right) \times V \tag{18}$$

where, $g_c$ (5.43 for piezoelectric crystals) is a potential output coefficient and s and t are the area and thickness of the sensor, respectively. For the accelerometer used in the experimental setup, the results for the acceleration sensor were obtained from the producer [36]. These values are shown in Table 3.

**Table 3.** Accelerometer reference and measured values on the experimental setup.

| Given Reference (g) | Output Voltage Reference (V) | Measured Voltage (V) |
|---|---|---|
| 0.1 | 2.5 | 2.56 |
| 0.3 | 2.8 | 2.79 |
| 0.8 | 3.1 | 3.16 |
| 1.1 | 3.5 | 3.63 |
| 1.5 | 4 | 4.07 |

It was observed that the data obtained from the sensor and the measured values and the graph created with the data obtained from the experimental setup were compatible. The force calculation table and displacement table applied on the sensor are calculated as shown in Table 4 [35].

**Table 4.** Accelerometer reference and measured values on the experimental setup (given the coefficient value of the MEMS accelerometer is 5 [37]).

| Potential (mm) | Calculation of F (N) | Measured Sensor Value (V) | F = 5 × V (N) |
|---|---|---|---|
| 9.11 | 15.89 | 2.56 | 12.80 |
| 10.29 | 18.05 | 2.79 | 13.95 |
| 11.62 | 20.45 | 3.16 | 15.80 |
| 13.13 | 23.31 | 3.63 | 18.15 |
| 14.70 | 26.11 | 4.07 | 20.35 |

Based on the ISO 4407 [38] standard and the AS4059 standard replacing SAE 1638 [39], the pollution level of a fluid is determined by the maximum number of particles. According to the SAE 1638 standards, a maximum of 12 pollution levels are defined, with 1,024,000 or

more particles considered to be dirty oil. Based on this standard, when we compare the number of particles in the system (Table 5), the oil in the system, whose pollution level is consciously increased, is rated as dirty when it reaches 25% to 48% according to the number of 5–15 mm particles.

**Table 5.** Number of particles found in the oil according to NAS 1638 standards.

| Impurity Level | Number of Particles | | | | |
|---|---|---|---|---|---|
| | 5–15 μm | 15–25 μm | 25–50 μm | 50–100 μm | >100 μm |
| 0% | 34,720 | 617 | 57 | 3 | 0 |
| 25% | 528,438 | 9329 | 162 | 5 | 1 |
| 50% | 1,398,764 | 20,177 | 606 | 34 | 3 |
| 75% | 20,981,460 | 396,712 | 9866 | 626 | 26 |
| 100% | 524,536,541 | 9,568,693 | 232,837 | 16,419 | 668 |

## 6. Conclusions

The impurity of hydraulic liquid is one of the most critical factors affecting performance for oily systems. In this study, the effect of pollution caused by cavitation on hydraulic pumps was investigated. Keeping the factors that change the operating regime of the pump constant, five different pollution levels were examined. It was observed that a frequency component between 14–15 kHz was formed as different components at these impurity levels, and it was observed that this component also increased in parallel with the oil impurity level. Another important result is that the components between 0–0.5 kHz, which can be called the operating nominal frequency of the pump, also increased with pollution.

When the mathematical expression of cavitation was analyzed in the ZGB cavitation model used in this study, and if fixed values were defined as the default values in the literature, it can be seen that the values causing cavitation changed with vapor density, liquid density, and vapor volume fraction. Cavitation, which causes oil impurity, increases the free density in the oil; that is, it changes these three values. As a result, these values change the impurity and thus the operating conditions of the engine.

As explained in the study, cavitation is a function of bubble formation with many effects and values such as velocity, internal pressure, external pressure, and surface of the pressure change regions of these bubbles. For this reason, the impurity of the liquid circulating in the system is as important as the cavitation. In the experimental studies, the impurity of the oil was gradually increased. Oil at 100% contamination level was taken in the scale containers and the waste oil was mixed with the total oil in the system tank at each stage. Each time the chamber was emptied and cleaned, the mixture was prepared again according to the appropriate contamination level to be tested at each stage.

The change in the pressure areas of the bubbles suddenly burst when the appropriate conditions occurred, increasing the pressure several hundredfold. This pressure change can be shown as follows:

This example shows one spherical pressure value evaluation with respect to time. Bubble implosion occurs at about 200 ms. Moreover, a corrosive effect begins and the bubble is displaced by the pieces it has removed from the surface. The results of the accredited laboratory tests of the oils used in the study are as follows:

According to these results, it was observed that the number of particles in the oil increased due to the cavitation caused by the bursting of the bubbles. Therefore, it can be said that the amount of vibration will increase as the force applied to the pump walls and solid surfaces at the connection points increases with the bursting of a large number of bubbles and the resulting jet effect.

In summary, this study expressed mathematically the relationship between cavitation and oil pollution, and it was observed that the amplitude of the frequency component increased in accordance with the mathematical equations in the experimental studies.

Here, the frequency component formed with the pollution was measured as $14.78 \pm 0.3$ kHz. It has already been stated in many publications in the literature that the components to be formed due to cavitation will be between 2–25 kHz.

**Author Contributions:** Research, Ö.Y. and Z.K.; experimental work, Ö.Y. and Z.K.; writing—original draft preparation, Ö.Y.; MATLAB code writing, Z.K.; supervision, M.A.; interpretation of experimental results, M.A. All authors have read and agreed to the published version of the manuscript.

**Funding:** This research is externally funded by Eksim Holding- Dicle EDAS R&D Center.

**Acknowledgments:** The authors would like to acknowledge the active support of this research by the Science Institute of Çukurova University.

**Conflicts of Interest:** The authors declare no conflict of interest.

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
