# Peer review of "Investigation of the Relationship between Vibration Signals Due to Oil Impurity and Cavitation Bubbles in Hydraulic Pumps"

_electronics, doi:10.3390/electronics11101549_

Round 1

Reviewer 1 Report

Broad comments. In general, the text is well structured and has defined topics, but there are some drawings regarding the motivation, impact, and innovation of the work. More specific the text as is looking more like a nice technical report rather than a research paper. Authors should pay attention to describing the added value of their effort.

Specific comments. Below you may find some comments for improvement:

1. The abstract should be descriptive of the method and results of the manuscript. Authors should consider refining the abstract such that the approach followed is part of a generic issue.

2. The same applies to the introduction. The authors are advised to refine this section such it addresses the following:

  • What is the research question on hand?
  • Importance of the issue (more or less is well described in the text)
  • Who else worked on the issue and what are the limitations/borders?
  • What is the approach of the present work?
  • What is the innovation of the approach?

3. The authors could consider adding a paragraph describing the structure of the paper at the end of section 1.

4. It is emphasized that the motivation and impact of the work, which is believed by the reviewer to be very important, should be clearly described. The novelty should be clearly stated and described.

5. The authors need to review the state of the art on the specific issue as well as on similar engineering topics.

6. What is the purpose of section 3? Authors should clarify better the use of mathematical modeling.

7. Is section 4 needed? The DFT method is well known and normally all readers are familiar with it. Authors should consider refining the section such that the specific methodology, selections, computational setup, assumptions, etc are provided.

8. In section 5 the authors provide the main experimental results of their work. While the description of the experimental setup is well structured, further justification on selections made should be provided (e.g. pressure at 80 bars, etc)

9. In addition, the outcome of the analysis should be better illustrated. 3D graphs are good for qualitative representation but provide limited information to the reader.

10. Comparative results with other experimental results (state of the art) or simulations/numerical results should be given.

11. Other than demonstrating the application of FFT on acquired signals, authors should justify the importance of the outcome. Also, means/methods of verification and validation of the results should be provided.

12. Conclusions should provide (other than the main outcomes) a brief overview of the work.

Author Response

Hello

Thank you very much for your review report. I am sending a report as a pdf with the corrections I have made in response to your suggestions.

Our publication shows that the pressures on solid surfaces caused by cavitation exert forces on solid surfaces, causing them to vibrate at nominal operating conditions. No comprehensive study has been found in the literature that directly relates cavitation formation to vibration.

Thank you very much again for the suggestions you have shown in our edits. I hope the revisions will please you as well.

Best regards. 

Reviewer 2 Report

Dear authors,

The task to put in evidence the effect of cavitation to the operation of oil pump is very important in fault diagnosis and maintenance of hydraulic systems. Unfortunately, your paper does not convince me about the results you made. You should do a maximum revision of the paper.

Some suggestions I can add:

Title: It should be improved …

Abstract

r.11 - It is not clearly if the subject is on “lubricating oil” or is about “oil pumps”! Please, give a right description of the hydraulic system and the objective of research.

r.14 – energy transition? Or transmission?

r.17 – incomplete sentence, and few data are presented on the results of the experimental research

Keywords

r.20 – there are words which are not found in title, abstract and text (fault detection) or have low frequency (condition monitoring). The keyword list needs to be reviewed.

1..Introduction

A systematization of diagnostic methods and methods of signal processing in the case of vibration measurement should be done, together with the analysis of the existing literature.

r.56-58 – it should be reformulated ..

The purpose of the research and the steps that will be taken in this research should be described.

  1. Effect of Dirty Oil

The title of this section suggests that the effect of contaminants on the operation of the hydraulic system will be addressed.

Therefore, the text from r.93 to r.132 could be narrowed down and could be described more simply by a diagram or figure.

r.75 – heat is chemical contaminant???

r.98- water is vital or fatal..?

  1. Mathematical formulation of cavitation

The construction of the cavitation model is not convincingly presented and not all the factors that determine the cavitation are properly highlighted. The title of this section could be Factors influencing cavitation as a source of vibrations.

r.164-167- therefore, the 163 higher the oil impurity … it should be reformulated

r.191-194- in rel (3) and (4) the significance of n0 is the same???

r.198-200 – it should specify the significance of Re , Rc.

  1. Experimental Studies

A description of all components of the experimental setup and of the measurement procedure is required.

The mounting of the acceleration sensors must be described.

Some explanations about how the oil was purified are needed and what the 25%, 50%... contaminants mean.

r.299 – delimitations … or limitations?

In any case, the proposed experimental setup does not include all the elements that would allow obtaining conclusive results.

Check English

Author Response

(The authors gave the same response as above.)

Round 2

Reviewer 1 Report

The authors have covered previous suggestions. 

I would suggest refining the work once again such that you minimize the part of it that could be found in an oil flow related book. 
In addition, I would suggest to emphasize a bit ore the novelty of your method. 
The abstract could also be refined. While the inserted part is descriptive of the generic problem, the authors could consider adding some statistics related to the research question. 

Author Response

Dear reviewer,

The summary document of the actions I have taken after reviewing your suggestions is attached. I have also checked the spell check. 

Thanks again for your review.

Reviewer 2 Report

Dear authors.

The second version of your paper is much improved and easy for readers to understand.

There are still small corrections to be made:

  1. 52: The motor drive system that provides …
  2. 374: In the figure, the pressure must be denoted by the lower letter - p, as used above
  3. 401 – 403: idem
  4. 438: w, but …
  5. 467: 1.1 kW
  6. 471: In the figure, the three-dimensional coordinate system - does not need to be shown
  7. 474: .. are held are made
  8. 487: .. 7 cm … 2 cm ..

The results and discussions are conclusive.

Author Response

Dear reviewer

The summary document of the actions I have taken after reviewing your suggestions is attached.

Thanks again for your review.
